# Rotational Curves of the Milky Way Galaxy and Andromeda Galaxy in Light of Vacuum Polarization around Eicheon [†]

**Sergey L. Cherkas** [1,*,‡] and **Vladimir L. Kalashnikov** [2,‡]

1 Institute for Nuclear Problems, Bobruiskaya 11, 220006 Minsk, Belarus
2 Department of Physics, Norwegian University of Science and Technology, 7491 Trondheim, Norway; vladimir.kalashnikov@ntnu.no
* Correspondence: sergey_cherkas@list.ru
† This paper is an extended version of "Dark Matter in the Milky Way Galaxy as the F-Type of Vacuum Polarization." from the Proceedings of the 2nd Electronic Conference on Universe, Online, 16 February–2 March 2023.
‡ These authors contributed equally to this work.

**Abstract:** Eicheon properties are discussed. It is shown that the eicheon surface allows setting a boundary condition for the vacuum polarization and obtaining a solution describing the dark matter tail in the Milky Way Galaxy. That is, the dark matter in the Milky Way Galaxy is explained as the F-type of vacuum polarization, which could be treated as dark radiation. The model presented is spherically symmetric, but a surface density of a baryonic galaxy disk is taken into account approximately by smearing the disk over a sphere. This allows the reproduction of the large distance shape of the Milky Way Galaxy rotational curve. Andromeda Galaxy's rotational curve is also discussed.

**Keywords:** eicheon; dark matter; vacuum polarization; rotational curve; galaxy nuclei

## 1. Introduction

Observation of the stellar orbits around the center of the Milky Way Galaxy [1,2], detecting the gravitational waves from the black hole/black hole and black hole/neutron star coalescence (e.g., see the catalog [3] for an overview), radio-astronomy observation of the "black hole shadows" in the centers of galaxies [4,5] are widely considered as the direct evidence of an extremely compact astrophysical object (ECO) existence with a radius of an order of the Schwarzschild one. The observable properties of such an object are well-described by an exact Schwarzschild (or, more precisely, Kerr) solution of the general relativity (GR) equations [6,7]. A principal question is whether the Schwarzschild solution interprets reality quite adequately. Indeed, there are a lot of theoretical attempts to describe ECO whose properties approach those of an ordinary GR black hole sufficiently far from the event horizon (so-called horizonless "exotic compact objects" [8]). Some of them are based on the modified theories of gravity[1]. Recently, ECOs without a horizon have been discussed intensively (e.g., [12–15]). A zoo of exotic ECO, such as bosonic stars [16], gravastars [17], and other exotic stars [18,19], was proposed and theoretically explored. Also, the approaches based on constructing the nonsingular black-hole metrics in the spacetimes of different dimensions were proposed (e.g., see [20,21]).

The question about the nature of ECO is also related to the need for dark matter (DM) to explain the galactic rotational curves [22–24]. In particular, the first observation of the DM density around the stellar-mass ECO appears [25]. It was conjectured that the primordial black holes could be considered the candidates to DM [26]. Besides, there is a plethora of DM candidates [27]. However, could we advance without extraordinary physics but only by taking a vacuum polarization into account correctly [28]? Conventional answer is "No" in the frame of the renormalization technique of quantum field theory on a curved background [29,30]. Still, this approach demands covariance of the mean

value of the energy-momentum tensor over the vacuum state [29]. This demand has no solid foundation because it is known that there is no vacuum state invariant relative to the general transformation of coordinates. On the contrary, an argument was put forward that the preferred conformally-unimodular metric (CUM) could describe a vacuum polarization and resolve the DM problem [28,31]. In this metric, a black hole as an object having a horizon is absent [32]. Its disappearance results from the coordinate transformation relating the Schwarzschild-type metric to CUM, which selects some shell over the horizon and draws it into a node. As a result, a point mass without a horizon arises in a CUM. It is an idealized picture. In reality, one must know the equation of the state of a substance forming such ECO (named "eicheon" [32]).

Here, aiming at understanding the eicheon nature, we will use an approximation of the constant energy density and a trial "equation of state" relating the maximal pressure and the energy density. In our approach, we construct eicheon, and, after determining its properties, describe an eicheon surrounded by "dark radiation" to explain the rotational curves of the Milky Way and Andromeda (M31) Galaxies. "Dark radiation" is one of two kinds of vacuum polarization considered in [28], namely the polarization of F-type. Finally, to be closer to observations, we introduce a baryonic matter into the model by smearing the galactic disks of the Milky Way Galaxy and M31.

## 2. What Is "Eicheon"?

Eicheon is a horizon-free object which appears instead of a black hole in CUM. As an idealized structure, eicheon represents a solution of a gravitational field of a point mass in CUM. In the metric of a Schwarzschild type, it looks like a massive shell situated over the Schwarzschild radius. In the real world, where there is no infinite density and pressure, the eicheon could be modeled in the Schwarzschild-type metric by a layer of finite width over the horizon, as it is shown in Figure 1. In CUM, it looks like a solid ball [32,33]. A constant density model is convenient for understanding the main features of the eicheon.

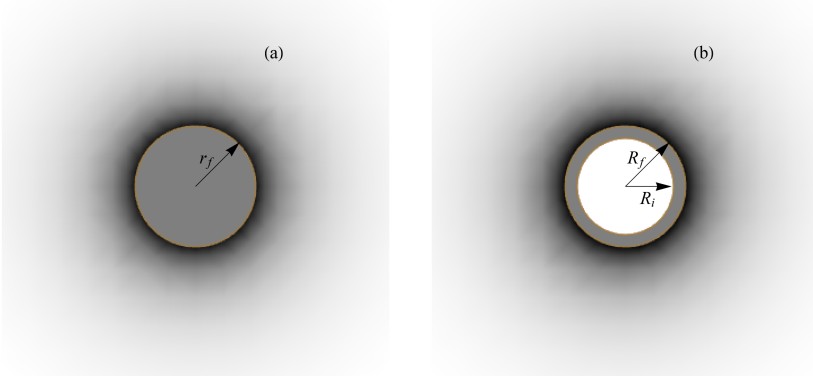

**Figure 1.** (**a**) Nonsingular eicheon surrounded by dark radiation in CUM (3) has a nonsingular core. (**b**) In the Schwarzschild type metric (4), this core looks like a hollow sphere. Vacuum polarization around an eicheon is shown as the gradient of a density.

CUM for a spherically symmetric space-time is written as

$$ds^2 = a^2(d\eta^2 - \tilde{\gamma}_{ij}dx^i dx^j) = e^{2\alpha}\left(d\eta^2 - e^{-2\lambda}(d\boldsymbol{x})^2 - (e^{4\lambda} - e^{-2\lambda})(\boldsymbol{x}d\boldsymbol{x})^2/r^2\right), \quad (1)$$

where $r = |\boldsymbol{x}|$, $a = \exp\alpha$, and $\lambda$ are the functions of $\eta, r$. The matrix $\tilde{\gamma}_{ij}$ with the unit determinant is expressed through $\lambda(\eta, r)$. The interval (1) could be also rewritten in the spherical coordinates:

$$x = r\sin\theta\cos\phi, \ \ y = r\sin\theta\sin\phi, \ \ z = r\cos\theta \quad (2)$$

resulting in

$$ds^2 = e^{2\alpha}\left(d\eta^2 - dr^2 e^{4\lambda} - e^{-2\lambda}r^2\left(d\theta^2 + \sin^2\theta d\phi^2\right)\right). \tag{3}$$

However, let us discuss eicheon properties in the Schwarzschild-type metric, which is more convenient for a reader

$$ds^2 = B(R)dt^2 - A(R)dR^2 - R^2 d\Omega. \tag{4}$$

In this metric, the Volkov-Tolman-Oppenheimer (TOV) equation for a layer $R \in \{R_i, R_f\}$ reads as:

$$p'(R) = -\frac{3}{4\pi M_p^2 R^2}\mathcal{M}(R)\rho(R)\left(1 + \frac{4\pi R^3 p(R)}{\mathcal{M}(R)}\right)\left(1 + \frac{p(R)}{\rho(R)}\right)\left(1 - \frac{3\mathcal{M}(R)}{2\pi M_p^2 R}\right)^{-1}, \tag{5}$$

where the function

$$\mathcal{M}(R) = 4\pi \int_{R_i}^{R} \rho(R')R'^2 dR' \tag{6}$$

and the reduced Planck mass $M_p = \sqrt{\frac{3}{4\pi G}} = 1.065 \times 10^{-8}$ kg. We will model a layer of constant density $\rho$ so that $\mathcal{M}(R)$ is expressed as

$$\mathcal{M}(R) = \frac{4\pi}{3}\rho\left(R^3 - R_i^3\right). \tag{7}$$

It is convenient to measure distances in units of the Schwarzschild radius $r_g = \frac{3\mathbb{M}}{2\pi M_p^2}$, while energy density and pressure in the units of $M_p^2 r_g^{-2}$. In these units, it follows from (7) and the definition of the eicheon mass $\mathbb{M} = \mathcal{M}(R_f)$ that

$$\rho = \frac{1}{2(R_f^3 - R_i^3)}. \tag{8}$$

The TOV Equation (5) is reduced to

$$p' = \frac{(p + \rho)\left(3pR^3 + \rho\left(R^3 - R_i^3\right)\right)}{R\left(2\rho\left(R^3 - R_i^3\right) - R\right)} \tag{9}$$

and has to be solved with the boundary condition $p(R_f) = p\left(\sqrt[3]{R_i^3 + \frac{1}{2\rho}}\right) = 0$, where the second equality follows from (8). Let us simplify the problem further and assume that $R_i = 1$ in the Schwarzschild radius units. Even in this case, there is no analytical solution of the Equation (9), but the most interesting quantity is a maximal pressure $p_{max} = p(1)$, which turns out to be approximated by the expression

$$p_{max} \approx \frac{\sqrt{\rho}}{\sqrt{6}} - \frac{1}{3} + \frac{11}{36\sqrt{6}\sqrt{\rho}} - \frac{35}{864\sqrt{6}\rho^{3/2}} \tag{10}$$

as is shown in Figure 2.

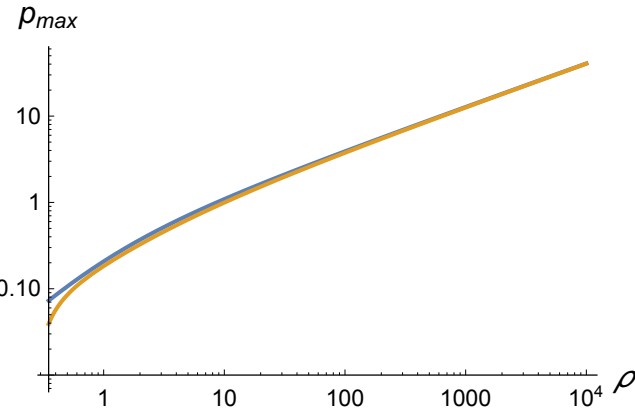

**Figure 2.** Pressure $p_{max}$ in the center of eicheon (see Figure 1a) in CUM, coinciding with the pressure $p(R_i)$ in the metric (4) (see Figure 1b). Blue and brown curves correspond to the numerical integration of the Equation (9) and approximation (10), respectively.

If one supplements Equation (10) by the "equation of state", which connects the maximal pressure with the density, then it is possible to determine the pressure and density. For instance, the "equation of state" corresponding to a degenerate relativistic fermion gas

$$p_{max} = \rho/3 \tag{11}$$

gives no solution because of Equations (10) and (11) are incompatible.

The equation of state of the nonrelativistic degenerate Fermi gas is written in physical units as [34]

$$\tilde{p}_{max} = \frac{1}{5}\left(\frac{3\pi^2}{m_N^4}\right)^{2/3}\tilde{\rho}^{5/3}, \tag{12}$$

where $m_N$ is a particle mass, and the tilde denotes that the quantity is expressed in the physical units. When $\rho$ is large, one could use only the first term in Equation (10), and its equating to the pressure from (12) gives the following expression

$$\frac{M_p r_g^{-1}\sqrt{\tilde{\rho}}}{\sqrt{6}} = \frac{3^{2/3}\pi^{4/3}}{5}\left(\frac{1}{m_N^4}\right)^{2/3}\tilde{\rho}^{5/3}, \tag{13}$$

allowing us to find the physical density

$$\tilde{\rho} = \frac{\left(\frac{5}{3}\right)^{6/7}2^{3/7}m_N^{16/7}M_p^{18/7}}{3\pi^{2/7}\mathbb{M}^{6/7}}, \tag{14}$$

which decreases with an increase of mass $\mathbb{M}$ of the eicheon. Dimensionless density is found by dividing (14) by $M_p^2 r_g^{-2}$ and reads

$$\rho = \frac{\sqrt[7]{3}\,5^{6/7}\mathbb{M}^{8/7}m_N^{16/7}}{2^{11/7}\pi^{16/7}M_p^{24/7}}. \tag{15}$$

It grows with the increase of $\mathbb{M}$, so that approximation $p_{max} \approx \sqrt{\rho/6}$ becomes justified at some mass according to (10). Respectively, the width of the eicheon shell decreases: $\Delta R = \sqrt[3]{1 + \frac{1}{2\rho}} - 1 \approx \frac{1}{6\rho}$ and becomes very thin at large $\mathbb{M}$. Certainly, we imply the relative width in units of $r_g$. For instance, if one takes the eicheon mass equal to the Sun mass $\mathbb{M} = M_\odot = 1.989 \times 10^{30}$ kg and $m_N$ equals the neutron mass, then the dimensional density $\tilde{\rho} = 2.4 \times 10^{19}$ kg/m$^3$, while the dimensionless $\rho$ equals 0.66. This eicheon has a rather thick skin $\Delta R \approx 0.33$ and, in the principle can be distinguished from a conventional black hole.

One more example is the eicheon of a large mass $40 \times 10^9 M_\odot$. In this case, the physical density is much lower and we could consider the "equation of state" for a cold hydrogen plasma, where the pressure is created by a degenerate electron gas, and the dimensional density satisfies

$$\frac{M_p r_g^{-1} \sqrt{\tilde{\rho}}}{\sqrt{6}} = \frac{3^{2/3} \pi^{4/3}}{5} \left(\frac{1}{m_e^4}\right)^{2/3} \left(\frac{\tilde{\rho}\, m_e}{m_N}\right)^{5/3}, \tag{16}$$

where $m_e$ is electron mass and $\tilde{\rho}\, m_e / m_N$ is electron density in a fully ionized hydrogen plasma. Thus

$$\tilde{\rho} = \frac{\left(\frac{5}{3}\right)^{6/7} 2^{3/7} m_e^{6/7} m_N^{10/7} M_p^{18/7}}{3 \pi^{2/7} \mathbb{M}^{6/7}}. \tag{17}$$

The dimensionless density is given by

$$\rho = \frac{\sqrt[7]{3}\, 5^{6/7} \mathbb{M}^{8/7} m_e^{6/7} m_N^{10/7}}{2^{11/7} \pi^{16/7} M_p^{24/7}}. \tag{18}$$

Numerically, these values are $\tilde{\rho} = 3.1 \times 10^7$ kg/m$^3$, $\rho = 1.4 \times 10^9$. The eicheon skin is very thin $\Delta R \sim \frac{1}{6\rho} \sim 10^{-10}$. Such eicheon is indistinguishable from a conventional black hole. At the same time, it is rather "mellow" by virtue of (17). Certainly, there is no paradox here because $\Delta R$ is measured in the units of $r_g$, which is large in the case considered. Finally, we can estimate eicheon in the center of the Milky Way Galaxy using the Formulas (16)–(18). For $\mathbb{M} = 4.154 \times 10^6 M_\odot$, they give $\rho \approx 3.8 \times 10^4$, $\Delta R \sim 10^{-6}$ and $\tilde{\rho} \approx 8.2 \times 10^{10}$ kg/m$^3$ that is greater than the white dwarf mean density $\tilde{\rho} \approx 4 \times 10^8$ kg/m$^3$ [35]. The eicheons of any mass exist because the inner $R_i$ and outer $R_f$ radii (see Figure 1b) exceed the Schwarzschild radius, and Buchdahl's bound [36] $\mathbb{M} < 4R/9G$ is not eligible.

To consider eicheon in CUM (3), one could set $t = \eta$, $R = R(r)$ and compare the metrics (3) and (4) to obtain:

$$B(R) = e^{2\alpha}, \tag{19}$$

$$R^2 = r^2 e^{-2\lambda + 2\alpha}, \tag{20}$$

$$A(R)\left(\frac{dR}{dr}\right)^2 = e^{4\lambda + 2\alpha}. \tag{21}$$

Using (19) and (20) in (21) to exclude $\lambda$ and $\alpha$ yields

$$\frac{dr}{dR} = \frac{R^2}{r^2} \frac{A^{1/2}}{B^{3/2}}. \tag{22}$$

In the region filled by matter, $A(R)$ and $B(R)$ obey [37]

$$\frac{d}{dR}\left(\frac{R}{A}\right) = 1 - 6\rho R^2,$$
$$\frac{1}{B}\frac{dB}{dR} = -\frac{2}{p + \rho}\frac{dp}{dR}. \tag{23}$$

For the model of a constant density $\rho(R) = const$, the Equation (23) can be integrated explicitly

$$A = \frac{R}{R - 1 - 2\rho\left(R^3 - R_f^3\right)}. \tag{24}$$

$$B = \left(1 - \frac{1}{R_f}\right)\frac{\rho^2}{(p(R) + \rho)^2} \approx 1 - \frac{1}{R_f}, \tag{25}$$

where the pressure is neglected compared to the energy density in the last equality of (25). According to (22), the eicheon radius is

$$r_f = \sqrt[3]{3 \int_1^{R_f} \frac{A^{1/2}}{B^{3/2}} R^2 dR} \approx \frac{\sqrt{3}\sqrt[3]{11}\rho^{1/6}}{2^{5/6}} + \frac{43}{2^{5/6} 3^{5/2} 11^{2/3} \rho^{5/6}}, \tag{26}$$

where a small "thickness" of the eicheon surface $R_f - 1$ is assumed, and $R_f$ is expressed as $R_f = \sqrt[3]{1 + \frac{1}{2\rho}}$. For a supermassive eicheon, using (18) and first term of (26) results

$$r_f \approx \frac{3^{11/21}\sqrt[7]{5}\sqrt[3]{11}}{2^{23/21}\pi^{8/21}} \sqrt[42]{\frac{\mathbb{M}^8 m_e^6 m_N^{10}}{M_p^{24}}}, \tag{27}$$

i.e., in CUM, the eicheon radius in the units of $r_g$ increases when the eicheon mass $\mathbb{M}$ rises.

## 3. Vacuum Polarization around of Eicheon

Considering the vacuum polarization for an arbitrary curved space-time background is a highly complex problem. Instead, one could consider the scalar perturbations of CUM:

$$ds^2 = (1 + \Phi(\eta, \boldsymbol{x}))^2 \left( d\eta^2 - \left( \left( 1 + \frac{1}{3}\sum_{m=1}^{3} \partial_m^2 F(\eta, \boldsymbol{x}) \right) \delta_{ij} - \partial_i \partial_j F(\eta, \boldsymbol{x}) \right) dx^i dx^j \right) \tag{28}$$

and calculate a spatially nonuniform energy density and pressure arising due to vacuum polarization in the eikonal approximation [28].

As was shown [28], the energy density and pressure of vacuum polarization corresponding to the F-type of metric perturbations (28) have the radiation equation of state $\delta p_F = \frac{1}{3}\delta\rho_F$. "Dark radiation" does not consist of real particles[2], nor interacts with some substance, but could be a source in the equations for gravity. That gives a possibility to use a hypothetical "dark radiation" in some heuristic nonlinear models, such as the TOV equation. For a radiation substance alone, a singular solution of the TOV equation exists that is devoid of physical meaning [37]. However, the situation changes cardinally in CUM in the presence of the nonsingular eicheon. This gives a possibility to set a boundary condition for a radiation fluid at $r = 0$ and obtain a nonsingular solution, including the dark radiation. In the Schwarzschild type metric (4), the boundary condition is set at the radial coordinate of an inner shell $R = R_i$, which corresponds to the point $r = 0$ in CUM (see Figure 1).

The system of equations (see Appendix A) in the metric (4), implies three substances: the eicheon of the constant density $\rho_1$, the dark radiation density $\rho_2$, and the density $\rho_3$ of baryonic matter of the galactic disk and bulge:

$$\begin{cases} \begin{cases} p_1' = -\frac{3(p_1 + \rho_1)\left(\mathcal{M} + 4\pi R^3\left(p_1 + \frac{\rho_2}{4}\right)\right)}{2R(2\pi R - 3\mathcal{M})}, & \mathcal{M}' = 4\pi R^2(\rho_1 + \rho_2), \\ \rho_2' = -\frac{6\rho_2\left(\mathcal{M} + 4\pi R^3\left(p_1 + \frac{\rho_2}{3}\right)\right)}{R(2\pi R - 3\mathcal{M})}, \end{cases} & R_i < R < R_f, \\ \rho_2' = -\frac{6\rho_2\left(\mathcal{M} + 4\pi R^3 \frac{\rho_2}{3}\right)}{R(2\pi R - 3\mathcal{M})}, & \mathcal{M}' = 4\pi R^2(\rho_2 + \rho_3), \quad R > R_f. \end{cases} \tag{29}$$

where the baryonic matter $\rho_3$ is considered as some external matter density. According to (29), there are two equations for the pressure and "dark radiation" density inside the eicheon and a single equation for "dark radiation" density outside the eicheon.

As is shown in the upper panel of Figure 3, the eicheon without galactic disk and bulge contributes at a small distance, and the dark radiation contributes at large distances. The density of dark radiation depends on the eicheon structure, which was considered in the previous section. It is convenient to introduce a universal quantity of a dark radiation density for the Milky Way Galaxy at the radius of a photon sphere $R = 3/2$, which almost does not depend on eicheon structure, namely $\rho_2^* \equiv \rho_2(3/2) = 9.3 \times 10^{-32} =$

$2.1 \times 10^{-25}$ kg/m$^3$. Moreover, it remains a single parameter because the eicheon mass in the dimensionless units equals $\mathbb{M} = 2\pi/3$. Thus, a DM tail is reproduced by virtue of the universal equations

$$\rho_2' = -\frac{3\rho_2\left(\mathcal{M} + 4\pi R^3\frac{\rho_2}{3}\right)}{\pi R\left(R - \frac{3\mathcal{M}}{2\pi}\right)}, \quad \mathcal{M}'(R) = 4\pi R^2\rho_2, \quad \mathcal{M}(3/2) = 2\pi/3, \quad \rho_2(3/2) = \rho_2^*, \quad R > 3/2. \tag{30}$$

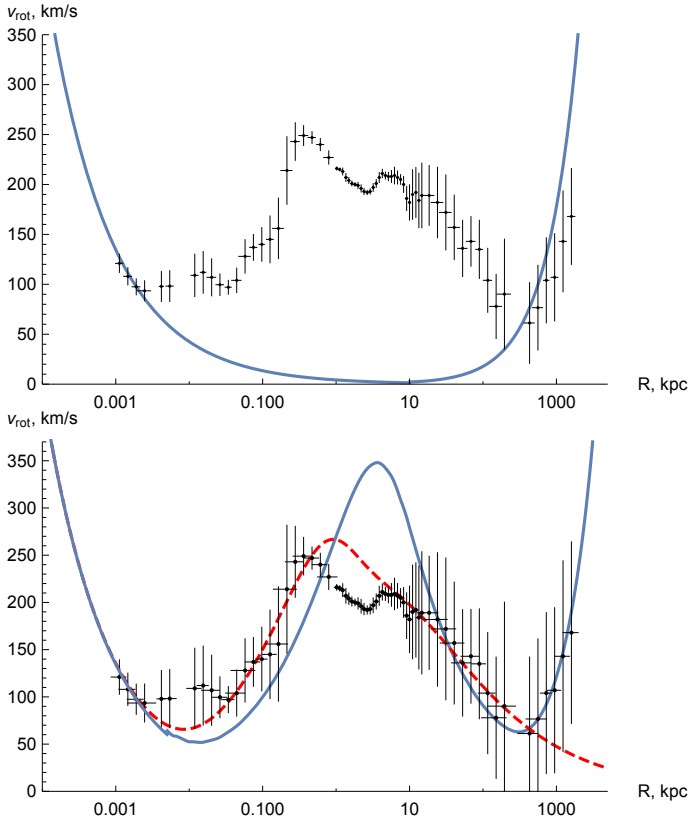

**Figure 3.** (**Upper panel**) The calculated rotational curve of the Milky Way Galaxy from Ref. [28], which includes contributions of the eicheon and dark radiation. (**Lower panel**) Rotational curve taking into account the baryonic matter by (33)–(35). Dashed line represents DM modelling by NFW profile, and baryonic matter by (33)–(35), but using another parameters. The result of observations with the error bars are taken from Ref. [22].

That is a spherically symmetric model where the amount of dark radiation is adjusted to fit the observations. The rotation velocity is calculated according to [28]

$$v_{rot} = \sqrt{\frac{R}{2B}\frac{dB}{dR}} = \sqrt{-\frac{R}{p_2 + \rho_2}\frac{dp_2}{dR}} = \frac{1}{2}\sqrt{-\frac{R}{\rho_2}\frac{d\rho_2}{dR}}, \tag{31}$$

where the last equality of (31) says that the dark radiation $\rho_2$ serves a "reference fluid" because satisfies continuity Equation (23) rewritten in the form of

$$\frac{d\rho_2}{dR} + \frac{2}{B}\frac{dB}{dR}\rho_2 = 0. \tag{32}$$

To consider the baryonic matter, one could smear a baryonic galactic disk on a sphere and view the resulting mass density as some external non-dynamical density in the TOV equations for the eicheon and dark radiation. This external density creates an additional gravitational potential.

Let us consider the surface density of matter in a galactic disk:

$$\wp = \frac{M_D}{2\pi R_D^2} e^{-R/R_D}, \tag{33}$$

and write the mass $dM$ corresponding to the radial distance $dR$

$$dM = \frac{M_D}{R_D^2} e^{-R/R_D} R \, dR = \frac{M_D}{R_D^2 R} e^{-R/R_D} R^2 dR. \tag{34}$$

According to (34), the smeared 3-dimensional density has the form:

$$\rho_3 = \frac{M_D}{4\pi R_D^2 R} e^{-R/R_D}. \tag{35}$$

The result of the calculations for the Milky Way Galaxy rotational curve is shown in the lower panel of Figure 3. As one can see, the simple model with smeared disk describes the baryonic matter roughly, but the observed rotational curve has a more complicated structure.

## 4. Andromeda Galaxy

Andromeda Galaxy (M31) is nearest to the Milky Way Galaxy and is situated at a distance $\sim$800 kpc. For M31, there are no small distances data $\sim$0.01 kpc, allowing us to identify a compact object in the center explicitly. Indeed, the situation is more complicated because cluster B023–G078 of M31 hosts one more black hole $\sim 10^5 \, M_\odot$ [38,39]. For the correct description, we need to apply a solution with two eicheons. This problem seems complicated, and we leave it for the future, considering only one central eicheon with the mass $10^8 \, M_\odot$ [38,39]. Calculations are analogous to that of the previous section.

The results of modeling are shown in Figure 4. The dimensionless parameter equaled the dark radiation density in the units of $M_p^2 \, r_g^{-2}$ at a photon sphere radius is $\rho_2(3/2) = 6.2 \times 10^{-28} = 2.3 \times 10^{-24}$ kg/m$^3$. This value is greater than that for the Milky Way Galaxy, i.e., these suggest the greater mass of central eicheon and the greater dark radiation density at a radius of the eicheon photon sphere. After introducing the baryonic matter by smearing galactic disk (33) over a sphere we have the curve shown in a lower panel of Figure 4.

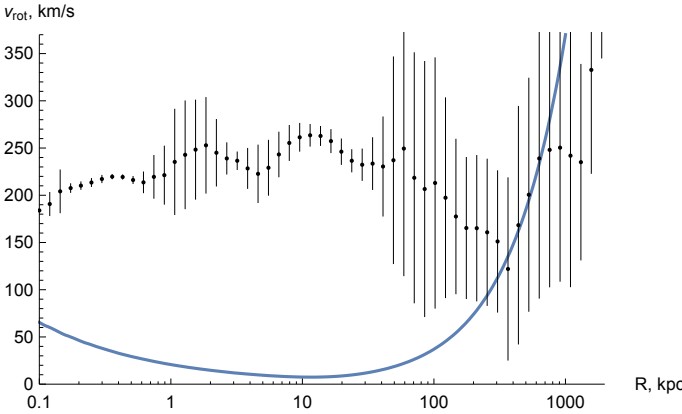

**Figure 4.** *Cont.*

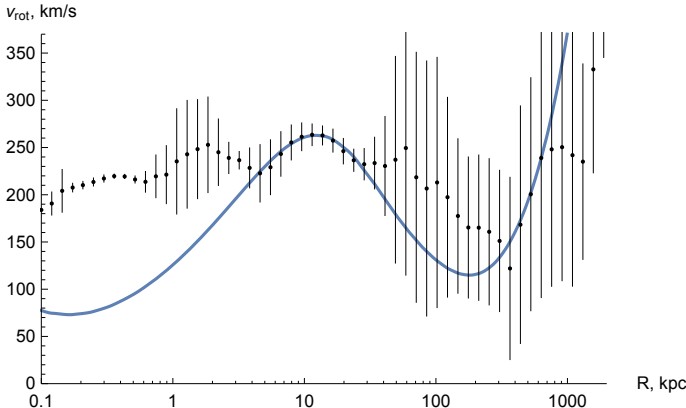

**Figure 4.** (**Upper panel**) Andromeda Galaxy rotational curve, which includes contributions of the eicheon and dark radiation only. (**Lower panel**) Rotational curve taking into account the baryonic matter by (33)–(35). The result of observations with the deviations bars are taken from Ref. [40].

## 5. Discussion and Conclusions

We have shown that the F-type vacuum polarization could explain DM, which mimics a sort of "dark radiation". Namely the presence of ECO, or *eicheon*, in the center of the galaxy provides a nonsingular solution for dark radiation. The eicheon resembles a black hole for an external observer but has no horizon. Our model is spherically symmetric. However, the appropriate approximation of the distribution of baryonic matter in a galaxy by the disk smearing over a sphere allows for obtaining the qualitative agreement of the rotational curves with the observed ones.

Still, in the spreading of a galactic disk, we overestimated a baryonic matter. Usually, it is supposed that DM begins to play a role from a few kpc, but according to the above consideration, the contribution of dark radiation becomes considerable only at tens kpc. For example, in Figure 3 (red dashed curve) we represent DM modelling by Navarro Frenk White (NFW) profile [41], where left and right edges of the central fold of the plot are approached by baryonic matter and NFW profile respectively.

For M31, we are not able to obtain an amount of DM needed in the region of 10–100 kpc. We conjecture that, if there is not only a central eicheon in the galaxy but a number of eicheons, one could glue dark radiation tails to every eicheons and create a needed amount.

Let us remind the principles of calculation. We have considered the vacuum polarization of F-type in CUM (28) and find that it has a radiation-like equation of state. Then, we solve the TOV equation for incompressible fluid and dark radiation and obtain a nonsingular solution. To consider the baryonic matter, we smear a galactic disk and use the resulting density as some external density. Interestingly, each galaxy's dark radiation tail can be described by a single parameter: density of dark radiation at the radius of a photon sphere of the eicheon. The numerical value of this density for the Milky Way Galaxy is $2.1 \times 10^{-25}$ kg/m$^3$, and for Andromeda Galaxy it is $2.3 \times 10^{-24}$ kg/m$^3$.

These values could be compared with the spatially uniform residual energy density of vacuum fluctuations, which remain after compensation of its main part by the constant in the Friedman equation [42]. It is of the order of critical density $\sim \times 10^{-26}$ kg/m$^3$. Certainly, we use the amount of the dark radiation at a photon radius of the eicheon $R = 3/2r_g$. Still, this amount rapidly decreases at $R > 3/2r_g$ and increases at $R < 3/2r_g$. In this light, it is interesting to obtain a general picture of matter structure formation in the universe by the solution of the system of the equations for the perturbations of the metric and the matter, including vacuum polarisation of both types [28].

**Author Contributions:** Concepts and methodology are developed by S.L.C. and V.L.K.; software, S.L.C.; validation, writing and editing, S.L.C. and V.L.K. All authors have read and agreed to the published version of the manuscript.

**Funding:** This research received no external funding.

**Data Availability Statement:** Not applicable.

**Conflicts of Interest:** The authors declare no conflict of interest.

## Appendix A. TOV Equation for a Mixture of Ordinary and Dark Fluids

Each of the fluids obeys the equation of the hydro-static equilibrium [37]:

$$\frac{B'}{B} = -\frac{2p_1'}{\rho_1 + p_1},\tag{A1}$$

whereas the equations for gravitational field give [37]

$$\left(\frac{R}{A}\right)' = 1 - 8\pi G \rho R^2,\tag{A2}$$

$$-1 + \frac{R}{2A}\left(-\frac{A'}{A} + \frac{B'}{B}\right) + \frac{1}{A} = -4\pi G(\rho - p)R^2,\tag{A3}$$

where $p = p_1 + p_2$, and $\rho = \rho_1 + \rho_2$. Solution of the Equation (A2) is written formally as

$$A = \frac{1}{1 - 2G\mathcal{M}/R},\tag{A4}$$

where $\mathcal{M}(R)$ is given by (6) Expressing $B'/B$ from (A1), $A$, $A'$ from (A4), (A3) and substituting them into (A2) gives

$$p_1' = -(p_1 + \rho_1)\frac{G(\mathcal{M} + 4\pi R^3 p)}{R(R - 2G\mathcal{M})}.\tag{A5}$$

The analogous equation holds for the second fluid.

## Notes

[1]   For the comprehensive reviews, see [9,10], and one may add an additional alternative approach based on the relativistic theory of gravity with a massive graviton [11].

[2]   One could imagine, that "dark radiation" consists of virtual particles of some kind. We have considered [28] only scalar field, minimally coupled with gravity.

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
