# Peer review of "Rotational Curves of the Milky Way Galaxy and Andromeda Galaxy in Light of Vacuum Polarization around Eicheon†"

_universe, doi:10.3390/universe9090424_

Round 1

Reviewer 1 Report

In this work the authors presents the so-called "Eicheons", solutions for the Conformally-unimodular metric, that comparing with a Schwarschild-type metric provides a object compact as a black-hole without an  event horizon . Considering a vacuum polarization F pertubation around this object to explain the rotation velocity of some galaxies.

First of all a more precise description of what "dark radiation " could means. . It a dark photon , an scalar particle?

1) in the definition of "Eicheon", paragraph 53, could be presented the reference of seminal work on this

2) On Eq. 16 , the density on Eq (13) is replaced by \tilde{\rho}m_e/M_N, without definition of m_e (Electron mass?) or why it appears.

3)the numerical procedure to obtain the solutions for Andromeda could be explicitly showed in the text.

Author Response

Authors are grateful to reviewer for the comments.

``Dark radiation'' is a notation of vacuum polarization of F-type. Because it is vacuum, it does not consist of real particles, nor interacts with some substance, but could be a source in equations for gravity. One could imagine, that ``dark radiation'' consists of virtual particles of some kind.  We have considered (Ref. [28]) only scalar field, minimally coupled with gravity.

We have include this note to the text of the paper, as well as the other notes, concerning referee's remarks.

Reviewer 2 Report

In this paper, the authors introduce "eicheon" as an alternative to black holes to explain the rotation curves of galaxies. Based on their previous research on the eicheon theory, they consider scalar perturbations in the conformally-unimodular metric for spacetime and calculate the density and pressure induced by the vacuum polarization around eicheon. The authors argue that the F-type vacuum polarization has a property corresponding to dark radiation, which can replace dark matter, thereby explaining the rotation curves of our Milky Way Galaxy. Using a combination of dark radiation from eicheon and a baryonic matter density distribution describing the Galactic disk, they predict galaxy rotation curves and compare them to observational data. They present an alternative model of dark matter that can reasonably account for observational data and offer scientifically meaningful insights. I believe that this paper contains scientific results that can be published in the journal Universe. Before recommending this paper for publication, we suggest that the authors revise the figures and text to clarify the following point. The authors compared their theoretical predictions with observations of the rotation curves of galaxies from small scales down to about 1000 kpc. With the help of baryon matter density, the vacuum polarization model provides a reasonable but not perfect description of the observations at scales below 10 kpc. On the other hand, at scales larger than 10 kpc, the model fits the observations well in both the Milky Way and Andromeda galaxies. Readers will want to know whether this result can only be explained by the vacuum polarization model or if it is easily explained by the introduction of general dark matter. Therefore, I think the authors should show how well the rotational velocity curves of galaxies at scales above 10 kpc are predicted using the dark matter halo model (e.g., NFW halo model), and compare these results with those of the vacuum polarization model. One more thing: the Milky Way should be written as the Milky Way Galaxy, and Andromeda should be written as the Andromeda Galaxy.

Author Response

Authors are grateful to reviewer for the comments.

In this version of the paper we have presented NFW modelling for the Milky Way Galaxy. NFW is a tools for precision modelling, which have to include budges, gas and dust. In our simple modelling we have combined NFW with the smeared galactic disk (with another parameters then in the eicheon modelling) only.